Plant traits and environment: floating leaf blade production and turnover of waterlilies

Klok Peter F. 1 2
van der Velde Gerard g.vandervelde@science.ru.nl 1 3
1 Department of Animal Ecology and Physiology, Institute for Water and Wetland Research, Radboud University Nijmegen , Nijmegen , Netherlands
2 Department of Particle Physics, Institute for Mathematics, Astrophysics and Particle Physics, Radboud University Nijmegen , Nijmegen , Netherlands
3 Naturalis Biodiversity Center , Leiden , Netherlands
de Vere Natasha
Electronic publication date: 2017 Apr 27
Publication date: 2017
Volume: 5
Electronic Location ID: e3212
Received 2016 Aug 2; Accepted 2017 Mar 21
Copyright: ©2017 Klok and Van der Velde
Copyright year: 2017
Copyright holder: Klok and Van der Velde
License: This is an open access article distributed under the terms of the Creative Commons Attribution License, which permits unrestricted use, distribution, reproduction and adaptation in any medium and for any purpose provided that it is properly attributed. For attribution, the original author(s), title, publication source (PeerJ) and either DOI or URL of the article must be cited.
License URL: https://creativecommons.org/licenses/by/4.0/

Keywords: Floating leaf blade production, Floating leaf life span, Nymphaeaceae, Nymphaeid growth form, Phenotypic variation, Seasonal change, Turnover rate

Funding: The authors received no funding for this work. The field study was carried out by the second author at the Laboratory for Aquatic Ecology, Catholic University of Nijmegen and financed by the Faculty of Science in the so-called vf programs.

==============================
Floating leaf blades of waterlilies fulfill several functions in wetland ecosystems by production, decomposition and turnover as well as exchange processes. Production and turnover rates of floating leaf blades of three waterlily species, Nuphar lutea (L.) Sm., Nymphaea alba L. and Nymphaea candida Presl, were studied in three freshwater bodies, differing in trophic status, pH and alkalinity. Length and percentages of leaf loss of marked leaf blades were measured weekly during the growing season. Area and biomass were calculated based on leaf length and were used to calculate the turnover rate of floating leaf blades. Seasonal changes in floating leaf production showed that values decreased in the order: Nymphaea alba, Nuphar lutea, Nymphaea candida. The highest production was reached for Nuphar lutea and Nymphaea alba in alkaline, eutrophic water bodies. The production per leaf was relatively high for both species in the acid water body. Nymphaea candida showed a very short vegetation period and low turnover rates. The ratio Total potential leaf biomass/Maximum potential leaf biomass (P/Bmax) of the three species ranged from 1.35–2.25. The ratio Vegetation period (Period with floating leaves)/Mean leaf life span ranged from 2.94–4.63, the ratio Growth period (Period with appearance of new floating leaves)/Vegetation period from 0.53–0.73. The clear differences between Nymphaea candida versus Nuphar lutea and Nymphaea alba, may be due to adaptations of Nymphaea candida to an Euro-Siberic climate with short-lasting summer conditions.

Introduction

Aquatic macrophytes can be considered as the basic frame of wetland ecosystems, called macrophyte-dominated systems (Den Hartog & Van der Velde, 1988; Jeppesen, Søndergaard & Christoffersen, 1998). The nymphaeid macrophyte growth form has mainly floating leaves, flowers on or elevated above the water surface and roots in the sediment of shallow open waters or littoral borders (Luther, 1983; Van der Velde, 1981). Waterlilies (Nymphaeaceae) as characteristic nymphaeids form the base of nymphaeid-dominated systems (Van der Velde, 1980). In the littoral zonation waterlilies grow often in between submerged macrophytes and helophytes, but the number of studies are few compared to those of the latter two groups. However, in shallow freshwater lakes, waterlilies can cover large areas (e.g., Brock, Van der Velde & Van de Steeg, 1987; Zbikowski, Kobak & Zbikowska, 2010), where submerged plants disappeared by accumulation of organic matter on the sediment during succession, eutrophication and acidification (Arts et al., 1990; Wiik et al., 2015). Waterlilies are adapted to such conditions (Dacey & Klug, 1979; Dacey, 1980; Dacey, 1981; Smits et al., 1990). Important factors for functioning in the aquatic wetland ecosystem are patterns of floating leaf production, decomposition and turnover. Leaves have a life history as their pattern of behavior from development from a primordium on a meristem to death by senescence or by environmental conditions which cause the dying off of the living tissue. During its existence it changes from being an importer and consumer of resources to being an exporter (Harper, 1977). The development and senescence of the leaves seem to be controlled by the plant itself through hormonal control (Chabot & Hicks, 1982).

Floating leaf blades (laminae) are important for exchange processes such as photosynthesis (Smits et al., 1988; Snir, Gurevitz & Marcus, 2006), gas exchange (Dacey & Klug, 1979; Dacey, 1980; Dacey, 1981; Ribaudo et al., 2012) and chemical accumulation e.g., by hydropotes (Lavid et al., 2001; Javadi et al., 2010). They make a significant contribution to the detritus food chain by decomposition (Brock, 1985; Brock, Boon & Paffen, 1985; Brock et al., 1985; Kok & Van der Velde, 1991; Kok, 1993) and function subsequently as food for detritivores (Kok et al., 1992). Floating leaf blades also have other functions, viz. as substratum for organisms at and near the water surface (e.g., Van der Velde, 1980; Hafner, Jasprica & Caric, 2013), as isles in the open water for air-breathing animals (e.g., Van der Velde & Brock, 1980; Willmer, 1982; Van der Velde et al., 1985), as substratum and nutrient for fungi (Vergeer & Van der Velde, 1997; Kowalik, 2012), as food for specialized invertebrate herbivores (Gaevskaya, 1969; Brock & Van der Velde, 1983; Van der Velde, Kok & Van Vorstenbosch, 1989; Kok, Van der Velde & Landsbergen, 1990) and for vertebrates (Gaevskaya, 1969; Paillison & Marion, 2001).

This study focuses on the floating leaf blade production of Nuphar lutea (L.) Sm., Nymphaea alba L. and Nymphaea candida Presl. Floating leaf blades are mentioned further in the paper as leaves. The research questions for this basic study are: Are there differences or similarities in production, turnover and other leaf characteristics between these waterlily species and are these differences related to environmental conditions by phenotypic plasticity or can they be considered biological species traits? For that purpose these aspects were compared between two co-existing species per site and between the same species in different sites.

Materials and Methods

Sites

Field research took place in three freshwater bodies in The Netherlands: Haarsteegse Wiel (HW), Oude Waal (OW) and Voorste Goorven (VG); dense nearly mono-specific stands occurred at all three sites. Within those stands, plots of a fixed area of 1 m2 were laid out: three for Nuphar lutea (HW and OW, 1977; VG, 1988), two for Nymphaea alba (OW, 1977; VG, 1988) and one for Nymphaea candida (HW, 1977). The Haarsteegse Wiel (Province of Noord-Brabant; 51°43′05″N, 5°11′07″E) is an isolated eutrophic water body with low alkalinity that consists of two connected breakthrough ponds created by dike bursts along the River Meuse. During the summer period, in the deep lake stratification occurs. The bottom of the nymphaeid stands consists of sand and a sapropelium layer with increasing thickness towards the littoral border. The Oude Waal (Province of Gelderland; 51°51′13″N, 5°53′35″E) is a highly eutrophic, alkaline oxbow lake in the forelands of the River Waal. The depth during the growth season is shallow, except for three remnants of former breakthrough ponds. The water level is dependent on precipitation, upward seepage, overflow of the River Waal in winter and/or spring (which strongly influences water chemistry and quality), and evaporation. The bottom consists of clay and sand, covered by a sapropelium layer of varying thickness in the nymphaeid beds. The Voorste Goorven (Province of Noord-Brabant; 51°33′53″N, 5°12′26″E) is a shallow, oligotrophic, isolated, culturally acidified moorland pool, showing very low alkalinity values. The hydrology is mainly dependent on precipitation, upward seepage and evaporation. The lake has a poorly buffered sandy soil. For further characteristics of the investigated water bodies, see Table 1. Chemical characteristics were derived from Brock, Boon & Paffen (1985) and Kok, Van der Velde & Landsbergen (1990).

Table 1 Physico-chemical characteristics of the three investigated water bodies.

Chemical characteristics from Brock, Boon & Paffen (1985) and Kok, Van der Velde & Landsbergen (1990).

	Haarsteegse Wiel (HW)	Oude Waal (OW)	Voorste Goorven (VG)	
Area (ha)	18	25	5	
Depth (m)	17	1.5	2	
Water level fluctuations	Low	High	Low	
Stratification	Yes (summer, thermocline at 4–6m)	No	No	
Hydrology	Precipitation/evaporation	Precipitation/evaporation	Precipitation/evaporation	
	Seepage	Upward seepage	Upward seepage	
		River water overflow		
Wind and wave action	Low	Moderate	Moderate	
Bottom	Sand/sapropelium	Sand/clay/sapropelium	Sand/sapropelium	
Trophic status	Eutrophic	Highly eutrophic	Oligotrophic	
Chemical characteristics				
Alkalinity (meq L−1)	1.5	5.2	0.0–0.07	
pH	7.1–8.5	6.7–8.3	4.7–5.5	
Sampling programme	1977	1977	1988	
Plots: species, depth (m)	Nuphar lutea, 1.5	Nuphar lutea, 1.5	Nuphar lutea, 2	
	Nymphaea candida, 2.5	Nymphaea alba, 1.5	Nymphaea alba, 2	

Solar radiation, air and water temperature

Daily measurements of solar radiation and air temperature collected by the Royal Netherlands Meteorological Institute during the growing season (April–December) were converted to 7-day moving averages, to eliminate large fluctuations on consecutive days. Also decade-averaged values for the periods 1971–1980 and 1981–1990 were used to get an impression of the “standard” yearly pattern for those periods. Water temperature data were collected weekly in the plots by means of a mercury thermometer at a depth of 5 cm at HW and OW in 1977 as part of the standard data taking procedure.

Field data from the plots

To collect field data, six representative plots of 1 m2 were laid out in the centre of mono-specific stands, surveying one rhizome apex per plot. A non-destructive leaf-marking method was used to mark all floating leaves within a plot, which enabled data collection during the complete life-span of the leaves. A square perforated PVC tube frame, held approximately 15 cm below the water surface by cork floaters and anchored by four bricks, bordered a plot (Fig. 1). In this way the unrolling of floating leaves in the plot was not hindered and all leaves having their petioles within the frame were counted and measured. A leaf was considered still present as long as, after fragmentation, tissue of the lamina was connected to the petiole in the case of OW and HW. In VG the leaf was considered gone when it was completely decayed but not fragmented and sunk under the water surface. The time that a leaf was present at the water surface was considered to be the leaf life span. Terms used in other studies are leaf persistence (Brock et al., 1983) and longevity (Chabot & Hicks, 1982).

Figure 1 A plot of Nuphar lutea (L.) Sm. bordered by an anchored frame provided with cork floaters.

The insert shows a tagged floating leaf with the tag code clearly visible. Illustration by John Slippens.

Measurements and observations of all leaves within a plot took place weekly during the growing season. It included tagging newly unrolled leaves by numbered Rotex tapes fixed around the petiole just under the leaf (insert Fig. 1), counting the actual number of leaves, measuring leaf length in mm (from the leaf tip to a basal lobe tip) and visually estimating leaf loss as percentage of the potential leaf area of each leaf. During the whole growing season, undamaged leaves were harvested at random a few metres outside the plots at each location to measure length (mm), area (cm2) and biomass (Ash-Free Dry Weight in mg). The vegetation period was calculated as the number of days beginning with leaf emergence at the water surface within the plot and ending on the first day that no leaf was observed any longer. The growth period was calculated as the number of days at which new floating leaves appeared at the water surface.

Potential, actual and photosynthetic leaf area and leaf biomass

Since floating leaf loss occurs, even before a leaf unrolls, a distinction was made between potential, actual and photosynthetic leaf area. The potential area was defined as the area of an entirely intact leaf. The actual area was defined as the potential area minus the area that is missing (e.g., caused by fauna, dehydration of the leaf margin, mechanical damage, absence of leaf parts due to decay). The photosynthetic area is defined as the actual area minus yellow, brown or decayed areas, thus the green leaf area left. The same distinction is made for leaf biomass.

Analysis of leaf length in time

Each leaf blade was measured weekly. To analyze the leaf length in time, both a linear and a quadratic curve fitting model were applied to make general leaf length predictions during the growing season per species per site:

(1) L1′t=a1+b1t

(2) L2′t=a2+b2t+c2t2

where:

Li′t=predicted leaf length (cm) at timet

t=time (day)

ai,bi,ci=correlation coefficients.

Corrected begin and end lengths of all leaves were used for the determination of the correlation coefficients. Values were corrected to maximum lengths to reflect the potential values. Curve fitting was performed separately on begin and end lengths, using both the linear model, corresponding to a linear increase (Eq. (1)), and the quadratic model, corresponding to a length optimum during the growing season (Eq. (2)).

Correlation of leaf area and leaf biomass with leaf length

Potential area and biomass were calculated by correlation with the end leaf length, using quadratic regression equations (Van der Velde & Peelen-Bexkens, 1983). Biomass is given in grams ash-free dry weight (AFDW). Field data from randomly harvested undamaged, fully green leaves outside the plots were used to determine equation coefficients. The equations were then applied to the field data from the plots. Mathematically, the equations for potential area and biomass are described by: (3) AiL=ciL2

(4) Bi,jL=ci,jL2

where: AiL=potential leaf area of speciesiat lengthLcm2

Bi,jL=potential leaf biomass of speciesiin water bodyjat lengthL(g AFDW)

L=leaf length (cm)

ci,ci,j=correlation coefficients

i=speciesNupharlutea,Nymphaeaalba,Nymphaeacandida

j=investigated water body (HW, OW, VG).

The total potential area (leaf area index) and biomass on each sampling date of species i in plot j, are calculated by summation over the individual leaves:

(5) Atot,t=ciΣLn,t2=ΣAiL

(6) Btot,t=ci,jΣLn,t2=ΣBi,jL

where:

Atot,t=total potential leaf area at timetcm2

Btot,t=total potential leaf biomass at timet(g AFDW)

Ln,t=length of leafnat timet(cm)

ci,ci,j=correlation coefficients

i=speciesNupharlutea,Nymphaeaalba,Nymphaeacandida

j=investigated water body (OW, HW, VG)

The total annual production of potential area and biomass was calculated by summation, using the maximum lengths of all produced leaves per species per plot for conversion to biomass data with the aid of the regression equations between leaf length and biomass based on the leaves collected outside the plot. Actual and photosynthetic area and biomass were calculated from the potential area and biomass by subtracting the field data loss percentages per leaf per species per plot.

Turnover rates and other ratios

Ratios give information on the regulation of the floating leaf production by the plant itself irrespectable of or in response to environmental conditions and therefore can be considered species traits (Kok, Van der Velde & Landsbergen, 1990). The turnover rate of leaves is calculated as the ratio Vegetation period/Mean leaf life span (Brock et al., 1983) and as the ratio Total potential biomass production/Maximum potential biomass (P/Bmax). Other ratios used are Mean leaf life span/Total number of leaves, Growth period/Vegetation period and the ratios Maximum leaf area/Total leaf area, Maximum leaf biomass/Total leaf biomass and Leaf area/Leaf biomass for the potential, actual and photosynthetic maxima.

Statistics

The package R (R Core Team, 2015) was used for statistics. Mean values with standard deviations have been computed for leaf life span, potential begin- and end leaf length, potential leaf area and potential leaf biomass. Welch’s t-test was used to test for differences in leaf life span between species for all plots. Linear regression was used to test for differences in potential begin and end leaf length between species for all plots.

Results

Solar radiation, air and water temperature

The 1977 and 1988 solar radiation and air temperature patterns were sinusoidal-shaped curves with a yearly period (Fig. 2).

Figure 2 Above, scale left: water temperature measured in Haarsteegse Wiel (HW) and Oude Waal (OW) in 1977 and averaged air temperature in 1977 and 1988 based on data collected by the Royal Netherlands Meteorological Institute. Below, scale right: averaged solar radiation in 1977 and 1988 based on data collected by the Royal Netherlands Meteorological Institute.

Number of leaves, leaf life span and vegetation period

Nymphaea alba showed the highest floating leaf production, followed by Nuphar lutea, and Nymphaea candida. The production of the floating leaves is sequential without cohorts.

The curves of the cumulative total number of leaves show that leaf production stopped after the beginning of September for Nuphar lutea and Nymphaea alba and in the first half of August for Nymphaea candida (Fig. 3).

Figure 3 Number of leaves: seasonal patterns of the cumulative (grey line) and actual (black line) number of leaves of one rhizome apex per plot.

The period with actual number of leaves is the vegetation period, the period with cumulative number of leaves indicates the period with newly appeared leaves the growth period. Where (A) Nuphar lutea in Haarsteegse Wiel during 1977, (B) Nuphar lutea in Oude Waal during 1977, (C) Nuphar lutea in Voorste Goorven during 1988, (D) Nymphaea candida in Haarsteegse Wiel during 1977, (E) Nymphaea alba in Oude Waal during 1977, (F) Nymphaea alba in Voorste Goorven during 1988.

Maximum and mean leaf life span were highest for the Nuphar lutea and Nymphaea alba in the oligotrophic and acid VG and for Nymphaea candida in the eutrophic and alkaline HW (Table 2). Significant differences for the leaf life span were found between Nuphar lutea in the eutrophic, alkaline OW and in oligotrophic, acid VG, between Nuphar lutea in OW and Nymphaea alba in VG and between Nuphar lutea in OW and Nymphaea candida in HW (Table 3).

Table 2 Leaf characteristics of Nuphar lutea (L.) Sm., Nymphaea alba L. and Nymphaea candida Presl in Haarsteegse Wiel (HW), Oude Waal (OW) and Voorste Goorven (VG).

Species	Nuphar lutea	Nymphaea alba	Nymphaea candida	
Location	HW	OW	VG	OW	VG	HW	
Year	1977	1977	1988	1977	1988	1977	
Number of leaves								
Total	m−2 yr−1	77	59	22	108	80	43	
Maximum	m−2	36	28	10	50	34	31	
Mean new per day		0.39	0.34	0.12	0.60	0.44	0.32	
Date(s) of maximum		Aug. 2	Aug. 19	July 28	July 13	Aug. 11	Aug. 2	
			Aug. 25	Aug. 18	July 20			
					Aug. 11			
Leaf life span								
Maximum	d	86	73	91	77	84	92	
Minimum	d	7	12	7	7	7	7	
Mean	d	42.75	37.93	47.73	40.46	46.46	46.28	
Standard deviation	d	19.93	13.03	18.28	18.09	18.48	20.17	
Vegetation period								
Length	d	199	175	183	180	183	135	
Begin date		May 10	May 11	Apr. 28	May 11	Apr. 28	June 7	
End date		Nov. 24	Nov. 1	Oct. 27	Nov. 6	Oct. 27	Oct. 19	
Growth period								
Length	d	127	120	134	120	134	71	
Begin date		May 10	May 11	Apr. 28	May 11	Apr. 28	June 7	
End date		Sep. 13	Sep. 7	Sep. 8	Sep. 7	Sep. 8	Aug. 16	
Leaf length								
Maximum	cm	39.0	39.0	33.2	34.0	25.2	23.5	
Minimum	cm	22.0	14.0	14.5	5.5	7.3	7.5	
Range	cm	17.0	25.0	18.7	28.5	17.9	16.0	
Mean begin	cm	30.79	31.05	23.09	21.86	17.33	17.88	
Standard deviation	cm	2.48	4.21	4.97	6.44	3.19	2.69	
Mean end	cm	32.29	32.81	24.27	24.28	18.86	19.30	
Standard deviation	cm	2.68	4.33	5.38	7.11	3.32	3.04	
Range	cm	1.5	1.8	1.1	2.4	1.6	1.5	
Leaf area								
Tot. pot.	m2 m−2 yr−1	4.97	3.99	0.84	5.30	2.31	1.12	
Max. pot.	m2 m−2	2.39	1.93	0.39	2.79	1.03	0.83	
Max. act.	m2 m−2	1.98	1.54	0.37	2.34	1.00	0.77	
Max. phot.	m2 m−2	1.72	1.21	0.32	2.06	0.98	0.62	
Mean pot. per day	m2 m−2 d−1	0.0250	0.0228	0.0046	0.0295	0.0126	0.0083	
Mean pot. per leaf	m2 m−2	0.0645	0.0676	0.0384	0.0497	0.0288	0.0260	
Standard deviation	m2 m−2	0.0104	0.0158	0.0155	0.0239	0.0092	0.0075	
Max. pot. area date		Sep. 6	Aug. 19	July 28	Aug. 11	July 21	Aug. 2	
Max. act. area date		Aug. 2	Aug. 19	July 28	Aug. 11	July 14	Aug. 2	
Max. phot. area date		Aug. 16	Aug. 19	July 28	Aug. 11	July 14	July 19	
Leaf biomass								
Tot. pot.	g AFDW m−2 yr−1	447	245	111	348	312	96	
Max. pot.	g AFDW m−2	215	118	51	183	139	71	
Max. act.	g AFDW m−2	178	95	49	153	136	66	
Max. phot.	g AFDW m−2	155	74	43	135	133	54	
Mean pot. per day	g AFDW m−2 d−1	2.25	1.40	0.61	1.93	1.71	0.71	
Mean pot. per leaf	g AFDW m−2	5.81	4.15	5.06	3.22	3.91	2.24	
Standard deviation	g AFDW m−2	0.94	0.97	2.04	1.49	1.25	0.65	
Max. pot. biomass date		Sep. 6	Aug. 19	July 28	Aug. 11	July 21	Aug. 2	
Max. act. biomass date		Aug. 2	Aug. 19	July 28	Aug. 11	July 14	Aug. 2	
Max. phot. biomass date		Aug. 16	Aug. 19	July 28	Aug. 11	July 14	July 19	

Table 3 p-values of Welch’s t-test for the leaf life span between species.

	Nl HW	Nl OW	Nl VG	Na OW	Na VG	Nc HW	
Nl HW	–	0.09140	0.2775	0.4247	0.2289	0.359	
Nl OW		–	0.02849*	0.2994	0.001758**	0.02035*	
Nl VG			–	0.09918	0.7761	0.7719	
Na OW				–	0.02768*	0.1043	
Na VG					–	0.9606	
Nc HW						–	
Notes.

** 0.001 < p < 0.01

* 0.01 < p < 0.05

Nl Nuphar lutea

Na Nymphaea alba

Nc Nymphaea candida

HW Haarsteegse Wiel

OW Oude Waal

VG Voorste Goorven

Nymphaea candida clearly showed a short vegetation period, compared to Nuphar lutea and Nymphaea alba. Nuphar lutea in HW showed the longest vegetation period.

Leaf length in time

During the whole growing season, Nuphar lutea developed the longest leaves. The mean begin and end lengths of leaves of Nuphar lutea were relatively short in the oligotrophic, acid VG and for Nymphaea candida in the eutrophic, alkaline HW (Table 2). The leaf length highly varied for Nuphar lutea and Nymphaea alba in the eutrophic, alkaline OW (Fig. 4).

Figure 4 Scatter plots of observed begin and end leaf lengths over time and regression lines of predicted maximum values using quadratic regression equation (6).

(A–B) Nuphar lutea in Haarsteegse Wiel during 1977, (C–D) Nuphar lutea in Oude Waal during 1977, (E–F) Nuphar lutea in Voorste Goorven during 1988, (G–H) Nymphaea alba in Oude Waal during 1977, (I–J) Nymphaea alba in Voorste Goorven during 1988, (K–L) Nymphaea candida in Haarsteegse Wiel during 1977.

The mean begin and end lengths were high for Nuphar lutea in HW and OW, compared to Nymphaea candida in HW and Nuphar lutea and Nymphaea alba in VG. The mean leaf length growth was highest for Nymphaea alba in OW (Table 2).

The results of the analysis of the leaf length time patterns show better fits for the quadratic model (Eq. (2)) compared to the linear model (Eq. (1)), indicating an optimum in leaf length during the growing season (Table 4). The quadratic fit of Nuphar lutea in VG for only 22 end lengths, has a low correlation with no significance. High correlation and low significance levels are shown for Nymphaea alba (OW and VG, both begin and end lengths) and for Nymphaea candida (HW, begin length) and high correlation with higher significance are shown for Nuphar lutea (HW, OW, VG, both begin and end lengths, except VG end length). If the c-coefficient value of the quadratic equation is very low, the curve turns into a straight line. In this case the correlation constants and the significance values do not differ much, indicating that both the quadratic and the linear model can be used. This applies both to the all observed values dataset of Nuphar lutea (HW, end length) and to the end values dataset of Nuphar lutea (HW and OW), showing a tendency towards increasing leaf lengths during the growing season instead of the occurrence of an optimum.

Table 4 Leaf length in time regression equations (Eqs. (1) and (2)) with correlation coefficients for begin and end lengths.

See also Fig. 4.

Site	Year	N	B∕E	Linear model	r	p	Quadratic model	r	p	
Nuphar lutea								
HW	1977	77	B	L = 28.74 + 0.034t	0.53	<0.05	L = 20.00 + 0.225t − 0.0009t2	0.70	<0.05	
			E	L = 27.68 + 0.037t	0.63	<0.01	L = 22.55 + 0.115t − 0.0003t2	0.67	<0.01	
OW	1977	59	B	L = 29.10 + 0.042t	0.53	<0.05	L = 24.56 + 0.145t − 0.0005t2	0.57	<0.05	
			E	L = 30.73 + 0.033t	0.59	<0.01	L = 22.47 + 0.166t − 0.0005t2	0.68	<0.01	
VG	1988	22	B	L =  − 29.83 + 0.044t	−0.44	n.s.	L = 17.04 + 0.268t − 0.0016t2	0.72	<0.05	
			E	L =  − 31.31 + 0.024t	−0.37	n.s.	L = 22.94 + 0.106t + 0.0005t2	0.44	n.s.	
Nymphaea alba								
OW	1977	108	B	L = 24.48 + 0.022t	0.22	n.s.	L =  − 2.79 + 0.628t + 0.0030t2	0.93	<0.001	
			E	L = 23.36 + 0.035t	0.42	<0.05	L = 4.08 + 0.346t − 0.0011t2	0.72	<0.001	
VG	1988	80	B	L = 18.34 + 0.008t	0.11	n.s.	L = 4.11 + 0.362t − 0.0019t2	0.97	<0.001	
			E	L = 19.94 + 0.009t	0.18	n.s.	L = 4.70 + 0.281t − 0.0011t2	0.86	<0.001	
Nymphaea candida								
HW	1977	43	B	L = 20.68 + 0.021t	−0.18	n.s.	L =  − 46.47 + 1.300t − 0.0063t2	0.95	<0.001	
			E	L = 15.22 + 0.039t	0.68	<0.05	L = 2.37 + 0.223t − 0.0006t2	0.76	<0.05	
Notes.

HW Haarsteegse Wiel

OW Oude Waal

VG Voorste Goorven

N total number of leaves per m2 per year

B begin length of leaves

E end length of leaves

L leaf length (cm)

t time in days (where April 1 = 0)

r correlation

p significance level (n.s. = not significant)

For the potential begin leaf length, all plot data were significantly different with the exception of Nuphar lutea in the eutrophic, alkaline HW versus Nuphar lutea in the eutrophic, alkaline OW (p = 0.5763), Nymphaea alba in OW versus Nuphar lutea in oligotrophic, acid VG (p = 0.3040), and Nymphaea candida in HW versus Nymphaea alba in VG (p = 0.5400).

This was also found for the potential end leaf length with the exception of Nuphar lutea in HW versus Nuphar lutea in OW (p = 0.3200), Nymphaea alba in OW versus Nuphar lutea in VG (p = 0.9151), and Nymphaea candida in HW versus Nymphaea alba in VG (p = 0.6704) and idem for potential leaf area (p = 0.1687, 0.0290, 0.0893, resp.).

Length-area and length-biomass relations

Regression equations for the computation of leaf area and leaf biomass from end leaf length are shown in Tables 5 and 6. The ratio length/area ranged from high to low in the order Nymphaea alba, Nymphaea candida and Nuphar lutea. The ratio length/biomass ranged from high to low in the order Nymphaea alba, Nuphar lutea and Nymphaea candida.

Table 5 Length-area regression equations (Eq. (5)), used for the calculation of potential leaf area from leaf length (Van der Velde & Peelen-Bexkens, 1983).

Species	Equation	
Nuphar lutea	A = 0.623L2	
Nymphaea alba	A = 0.788L2	
Nymphaea candida	A = 0.695L2	
Notes.

A area

L leaf length (cm)

Table 6 Length-biomass regression equations (Eq. (6)), used for the calculation of potential leaf biomass from leaf length (Van der Velde & Peelen-Bexkens, 1983).

Species	Location, year	N	Equation	S.E.	r2	p	
Nuphar lutea	OW 1977	27	B = 0.00382L2	0.330	0.99	<0.001	
	HW 1977	10	B = 0.00561L2	0.324	0.99	<0.001	
	VG 1988		B = 0.00821L2				
Nymphaea alba	OW 1976/1977	84	B = 0.00510L2	0.339	0.98	<0.001	
	VG 1988		B = 0.01068L2				
Nymphaea candida	HW 1977	10	B = 0.00598L2	0.229	0.98	<0.001	
Notes.

B biomass

L leaf length (cm)

OW Oude Waal

HW Haarsteegse Wiel

VG Voorste Goorven

Leaf area and leaf biomass

The potential maxima were reached when relatively many large leaves were present. The dates of these maxima differed from the dates of the actual and photosynthetic maxima for Nuphar lutea and Nymphaea candida in the eutrophic, alkaline HW, and for Nymphaea alba in the oligotrophic, acid VG. Both mean potential leaf area and mean potential biomass (per day and per leaf) were low for the VG plots, as well as for Nymphaea candida, with the exception of the mean potential biomasses per leaf in VG, which were relatively high. So, although the total biomass production in VG was low, the mean production per leaf was relatively high. For Nymphaea candida the ratios of the maximum of potential, actual, photosynthetic and total potential for both leaf area and biomass clearly showed high values because of lower leaf loss (Table 7).

Table 7 Turnover rates and other ratios of leaf characteristics of Nuphar lutea (L.) Sm., Nymphaea alba L. and Nymphaea candida Presl in Haarsteegse Wiel (HW), Oude Waal (OW) and Voorste Goorven (VG).

See also Table 2.

Species	Nuphar lutea	Nymphaea alba	Nymphaea candida	
Location	HW	OW	VG	OW	VG	HW	
Year	1977	1977	1988	1977	1988	1977	
Turnover rate								
Vegetation period/Mean leaf life span		4.63	4.61	3.81	4.39	3.89	2.94	
Turnover rate (P/Bmax)								
Tot. pot. leaf biomass/Max. pot. leaf biomass	yr−1	2.08	2.08	2.18	1.90	2.25	1.35	
Total number of leaves/Max. number of leaves		2.14	2.11	2.20	2.16	2.35	1.39	
Mean leaf life span/Total number of leaves		0.56	0.64	2.18	0.38	0.56	1.07	
Growth period/Vegetation period		0.64	0.69	0.73	0.66	0.73	0.53	
Leaf area								
Max. pot. / Tot. pot.	%	48	48	46	53	46	74	
Max. act. / Tot. pot.	%	40	39	44	44	43	69	
Max. phot. / Tot. pot.	%	35	30	38	39	42	56	
Leaf biomass								
Max. pot. / Tot. pot.	%	48	48	46	53	45	74	
Max. act. / Tot. pot.	%	40	39	44	44	44	69	
Max. phot. / Tot. pot.	%	35	30	39	39	43	56	
Leaf area/Leaf biomass								
for Max pot.		0.011	0.016	0.008	0.015	0.007	0.012	
for Max act.		0.011	0.016	0.008	0.015	0.007	0.012	
for Max phot.		0.011	0.016	0.007	0.015	0.007	0.011	

Seasonal patterns of potential, actual and photosynthetic area and biomass show a rapid and steady decline in floating blade area for Nuphar lutea in the eutrophic, alkaline HW and for Nuphar lutea and Nymphaea alba in the eutrophic, alkaline OW, in contrast to minimal decline for Nymphaea candida in HW and Nuphar lutea and Nymphaea alba in the oligotrophic, acid VG. The leaf area and biomass patterns also showed differences between species and location. Nuphar lutea showed highest values in the eutrophic, alkaline HW, intermediate values in the eutrophic, alkaline OW and lowest values in the oligotrophic, acid VG. Similarly, Nymphaea alba showed higher values in OW and lower in VG. In HW the Nuphar lutea values were higher than for Nymphaea candida. In OW, as well as in VG, the Nymphaea alba values were higher than the Nuphar lutea values (Fig. 5).

Figure 5 Changes over time of potential, actual and photosynthetic leaf area and biomass.

(A) Nuphar lutea in Haarsteegse Wiel during 1977, (B) Nuphar lutea in Oude Waal during 1977, (C) Nuphar lutea in Voorste Goorven during 1988, (D) Nymphaea candida in Haarsteegse Wiel during 1977, (E) Nymphaea alba in Oude Waal during 1977, (F) Nymphaea alba in Voorste Goorven during 1988.

The potential biomass in nearly all plots appeared to be significantly different with the exception of Nuphar lutea in OW versus Nymphaea alba in VG (p = 0.2197).

Turnover rates and other ratios

The ratio Total/Maximum number of leaves was approximately constant for Nymphaea alba and Nuphar lutea, but was much lower for Nymphaea candida (Table 7).

The turnover rate Vegetation period/Mean leaf life span was high for Nuphar lutea in both eutrophic and alkaline HW and OW and Nymphaea alba in OW, lower for Nuphar lutea and Nymphaea alba in oligotrophic, acid VG and lowest for Nymphaea candida in HW (Table 7). The turnover rate P/Bmax showed similar values for Nuphar lutea and Nymphaea alba and a clearly lower value for Nymphaea candida. The other ratios showed similar trends. The ratio Mean leaf life span/Total leaf production however showed the highest value for Nuphar lutea in VG followed by Nymphaea candida in HW, and much lower values in the other plot (Table 7).

LAI (leaf area index or maximum potential leaf area) ranged in our study from 0.39–2.79 m2.m−2 (Table 2).

Discussion

The 1977 and 1988 solar radiation and air temperature patterns were in good agreement with the decade-averaged values from the Royal Netherlands Meteorological Institute (http://www.sciamachy-validation.org/climatology/daily_data/selection.cgi, accessed April 18, 2017, and http://www.knmi.nl/index_en.html, accessed June 22, 2014), indicating that 1977 and 1988 can be considered “normal” years. The radiation curves showed an optimum in June and the temperature curves showed an optimum in July-August, indicating a lag period of at least a month. After their optima, the radiation decrease was faster than the temperature decrease. Similar patterns were described by Howard-Williams (1978), Jacobs (1979) and Nienhuis & De Bree (1980). The water temperature was closely related to the air temperature, especially in the shallow OW and VG. The water temperature pattern in HW showed smaller fluctuations, due to the greater depth with, consequently, a more stable temperature balance. Besides, after the summer optima of radiation and temperature had passed, the water in HW stayed longer at higher temperature levels compared to OW and VG.

The curves reflecting the actual number of leaves show maxima around August, corresponding to the maxima of the temperature curves and not to the June maxima of the radiation curves which show a gradual decrease afterwards. Therefore, water temperature seems to be a limiting factor for leaf production and not radiation.

Acidity might be an important factor during leaf development. Lower pH values of VG influence the ratio between total and maximum number of leaves to higher values for Nymphaea alba and Nuphar lutea, compared to HW and OW. However, wind and wave action (low for HW) and trophic status (eutrophic, alkaline for HW, highly eutrophic and alkaline for OW and oligotrophic, acid for VG) might also be relevant factors.

The mean leaf life span of Nuphar lutea and Nymphaea alba in the nutrient poor VG was relatively long, which agrees with previous research (Brock, 1985). Differences in maximum and mean leaf life span of species are quite small. The range of the mean floating leaf life span in our study was 38–48 days. These values cannot be considered a differentiating species trait. In the literature we found a mean floating leaf life span of 31 days for Nuphar advena in the USA (Twilley et al., 1985), 40 days for Nuphar japonica in Japan (Aramaki, Tsuchiya & Iwaki, 1989) and 21–55 days for Nymphaea tetragona (Kunii & Aramaki, 1992).

The vegetation period of Nymphaea candida is definitely shorter than the periods of Nuphar lutea and Nymphaea alba, mainly because of a late start in June.

Low pH and oligotrophic environment of VG lead to low turnover rates compared to the alkaline and nutrient richer environments of HW and OW.

Leaf length appears to be influenced by the physico-chemical conditions in the case of Nuphar lutea and Nymphaea alba. The mean begin and end lengths of leaves were relatively short for the VG plots and for Nymphaea candida in HW. During the whole growing season, Nuphar lutea developed the longest leaves. The leaf length highly varied for Nymphaea alba in OW and to a lesser extent for Nuphar lutea in VG. This high variation can be attributed to the development of secondary rhizome short shoots that develop leaves with small lengths.

Seasonal patterns of the actual, potential and photosynthetic area and biomass clearly show a difference between initial decomposition in alkaline and acid waters: during most of the growing season the relative actual and photosynthetic leaf area and biomass ratios in the nutrient poor and acidified VG were stable in the range 100%–75% due to less leaf loss, while in the alkaline and eutrophic HW and OW, a rapid and steady decline started soon after leaf development. Therefore the nutrient poor, acid conditions of VG prevent leaf loss for Nuphar lutea and Nymphaea alba.

Leaf lengths and leaf growth in VG were relatively low for Nymphaea alba and Nuphar lutea. However, leaves of both species in VG do have relatively high biomasses. Evidently, low pH and low buffering capacity may lead to smaller leaves and relatively high biomass values per leaf. The low length-biomass regression curve of OW indicates relatively low leaf biomass values, compared to leaf area values. Hence, the Leaf length/Leaf biomass ratio was affected by environmental conditions.

Statistics show a clear difference for leaf life span between the species in the acidic and oligotrophic VG and Nuphar lutea in OW. Comparison of similar data (mean leaf life span, turnover rates) of other research shows Nymphaea candida as deviating species with respect to turnover rates.

Ribaudo et al. (2012) found LAI’s of 1.05 (0.17) and 1.19 (0.19) m2.m−2 for Nuphar lutea in N. Italy, Twilley et al. (1985) mentioned a LAI of 0.82 m2.m−2 for Nuphar advena, and Aramaki, Tsuchiya & Iwaki (1989) 1.5 m2.m−2 for Nuphar japonica which all fall within the range found in the present study.

The very low LAI and leaf production of Nuphar lutea in VG may be due to a lack of nutrients, in particular phosphate. This species occurs normally in eutrophic water with phosphate concentrations in water and interstitial water higher than for Nymphaea alba (Van der Velde, Custers & De Lyon, 1986). Hutchinson (1975) mentioned that the floating leaves of Nymphaea odorata contain less phosphorus than those of Nuphar advena when they occur sympatric in the same bodies of water, which may be due to genetic differences between species. We lack such information on the waterlily species studied by us. Kok, Van der Velde & Landsbergen (1990) showed that phosphorus concentrations in undamaged, recently developed marked floating leaves of Nymphaea alba as well as Nuphar lutea initially were much higher in OW than in VG. In the plots P and N of floating leaves of both species have been resorbed by the plant for 70–73% in OW and 61–69% in VG (Kok, Van der Velde & Landsbergen, 1990). This means that the waterlilies belong to the plants with the highest nutrient resorption efficiencies (Hemminga, Marbà & Stapel, 1999).

Conclusions

This study will answer the questions if there are differences or similarities in production, turnover and other leaf characteristics between waterlily species and if these differences are related to environmental conditions by phenotypic plasticity or can be considered biological species traits.

Water temperature seems to be a limiting factor for leaf production, but not radiation. Analysis of the leaf length time patterns show better fits for the quadratic model, indicating an optimum in leaf length during the growing season. Length-biomass regression equations are obviously influenced by environmental conditions such as low pH, low alkalinity and oligotrophic conditions versus high pH, high alkaline and eutrophic conditions.

In the case of low pH, low alkalinity and oligotrophy Nuphar lutea and Nymphaea alba showed heavier leaves (ratio Leaf length/Leaf biomass), a lower number of leaves, a higher leaf life span, a longer growth period, a lower leaf length and leaf area, and leaf biomass. These factors lead also to low floating leaf blade turnover rates (Vegetation period/Mean leaf life span) compared to those in alkaline and nutrient richer environments. However, P/Bmax differed not much between both species in the various plots and conditions, just as the vegetation and growth period. These can be considered to be regulated by the plants themselves. Nutrient poor, acid conditions prevent leaf loss by inhibiting decomposition, in contrast to decomposition under alkaline and eutrophic conditions where leaf area loss occurred continuous from the start (Kok, Van der Velde & Landsbergen, 1990).

Nymphaea candida clearly deviates in several traits from Nuphar lutea and Nymphaea alba which showed similar leaf characteristics and responses to environmental conditions. Nymphaea candida showed a lower ratio between total and maximum number of leaves, a shorter vegetation period, a lower turnover rate (Vegetation period/Mean life span) and a low P/Bmax. These differences may be caused by adaptation to northern and continental climate conditions, as this species has a very wide distribution from The Netherlands in the west to Siberia in the east but does not occur south of the Alps (Muntendam, Povel & Van der Velde, 1996). In contrast, Nymphaea alba occurs all over Europe, including south of the Alps, and Nuphar lutea is widely distributed from Europe to West Siberia, also south of the Alps. The climate zones of the species include roughly the temperate-boreal zone for Nymphaea candida and the meridional, submeridional and temperate zone for Nymphaea alba and Nuphar lutea (Meusel, Jäger & Weinhert, 1965).

Supplemental Information

Data S1 Field data of the sites Haarsteegse Wiel and Oude Waal in 1977

Click here for additional data file.

Data S2 Field data of the site Voorste Goorven in 1988

Click here for additional data file.

We thank M Ankersmid, R Kwak, R De Mooij, H Peeters, F Verhoeven, V Vintges and CJ Kok for collecting field data, HJWJ van Huet for help with data modeling and E Jongejans for help with statistics, J Slippens † (Illustration Department, Radboud University) for making Fig. 1, and the reviewers Gudrun Bornette, Marina Suzuki and one anonymous reviewer for their constructive remarks.

Additional Information and Declarations

Competing Interests

Author Contributions

Data Availability

The authors declare there are no competing interests.

Peter F. Klok analyzed the data, contributed reagents/materials/analysis tools, wrote the paper, prepared figures and/or tables, reviewed drafts of the paper.

Gerard van der Velde conceived and designed the experiments, performed the experiments, analyzed the data, wrote the paper, prepared figures and/or tables, reviewed drafts of the paper.

The following information was supplied regarding data availability:

The raw data has been supplied as a Supplementary File.

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
