# Peer review of "Plant traits and environment: floating leaf blade production and turnover of waterlilies"

_PeerJ, doi:10.7717/peerj.3212_

## Round 0.1 · original submission · Minor Revisions

Dear Dr. van der Velde,

We have finally received the reports from three reviewers on your manuscript “Plant traits and environment: floating leaf blade production and turnover of waterlilies”. See below the comments provided by the reviewers. I am pleased to inform you that, based on the advice received, your manuscript may be considered acceptable for publication in PeerJ after a minor review is performed. Please, consider all the referees’ remarks in your revised manuscript and indicate in your rebuttal letter a point-by-point response to the referees.

With kind regards,

Ronaldo

Reviewer 1 ·

Basic reporting

No Comments.

Experimental design

Authors need to explain in more detail how the statistical analysis was performed. View comments in the PDF file.

Validity of the findings

I suggest improving the conclusions. View comments in the PDF file.

Additional comments

The manuscript presents important information about waterlies, an important group of macrophytes. The manuscript needs some fixes that are pointed in the PDF file. I am assent to its publication.

Annotated reviews are not available for download in order to protect the identity of reviewers who chose to remain anonymous.

·

Basic reporting

No comments

Experimental design

As is not possible to re-collect data, it would be recommended justify the only one quadrat by monospecific stand.

Validity of the findings

No comments

Additional comments

Despite the interesting results presented, a more current approach, for example, about the relationship between global climate changes and the distribution and development of the studied waterlilies species could be inserted.

·

Basic reporting

Abstract:
the abstract lacks some explanation about the acronym significance (B max, P) and the traits used (vegetation period : duration of the period between the first leaf production and the last leaf decay ?. And then, what is the difference between growth period and vegetation period? The percentage of damage is not discussed as a result. "Deviating climate" is perhaps not an accurate term. Euro-siberic climate is enough

Key-words:
I am not sure that phenotypic plasticity is accurate as a key word, as elements concerning phenotypic plasticity are not considered in the abstract. Furthermore, the variation of individual traits within species is done comparing different populations, and may not be due to phenotypic plasticity sensu stricto, but to population differentiation also. I suggest deleting this key word, and instead speaking about phenotypic variation.

Introduction:
the bibliographic references may be updated and much more rooted in the international literature.
Line 83-84: “The present study forms the base of studies on herbivory, various types of decomposition carried out in the same plots in the same wetlands”: I don’t feel this sentence is necessary, and it is confusing

Methods:
Line 124: the number of plots sampled is not indicated. How many rhizomes were surveyed per species? the indication should also be indicated on figure 3Line 156: it is not clear how the authors calculated total leaf biomass and biomass production, as 1) the leaves of a given rhizome were not collected, and 2) the way production was calculated was not explicated. Furthermore, I ask myself whether there is confusion between production (quantity of plant at a given sampling date) and productivity (production rate)
The paragraph dealing with the extrapolation of leaf biomass from leaf length should be positioned before the paragraph dealing with production and productivity. After this paragraph, the production is clear, but not the difference between leaf total biomass and production.
The origin of data informing about water quality must be given in this paragraph. There is a table with some data, which are not (or I missed something) introduced in the section.

Results
The result section is rather difficult to read, and may be more fluent if clear sub-sections referring to plant strategies, rather to traits, are identified. In the same way, the reference to abbreviated sites makes the reading very difficult, as it is complicated to remember which site is the oligotrophic, acidic or alkaline one. The site name may be replaced by their ecological characteristics : eg O_Alk, E_Aci, for eutrophication and alkalinity.

Discussion
Line 336-337 : there are an plenty papers dealing with the relationships between leaf life span and eutrophication, it may be good here to quote such papers.
Lines 355-368 : this may be moved to the results section, and the interest of the analysis more clearly explained in the method section.
Lines 374-375 : this may be due to a better leaf quality and life span for leaves produced in nutrient poor conditions. Several papers are outlining such patterns
Lines 376-382 : such contrast in regression slopes must be tested in an appropriate way (ancova)
Lines 400-402 : Nuphar lutea is related to standing, undisturbed waters, but is rather tolerant to contrasting trophic levels. The fact is that standing undisturbed waters are associated frequently with organic matter accumulation, and to nutrient enrichment (P and N)

Conclusion:
the conclusion is rather disappointing, as it only describe broadly the niche of species, but don’t go further. The consequences of such niches and phenotypic variation for species fitness, and the consequences of climate change on such niches, may be a good issue of this study, which is not considered in the conclusion.

Experimental design

the paper fits in journal scope
the research question is meaningful
the investigation is OK, the methods shoud be improved (see comments above)

Validity of the findings

the statistics may me improved in some way, and simplified.
the conclusion may be enlarged to the potential impacts of the findings

---

## Round 0.2 · Minor Revisions

Please pay consideration to the final suggestion from Reviewer 3 who suggests that reference to damage should be removed throughout

·

Basic reporting

The MS presented a qualitative improvement due to the other reviewers comments.

Experimental design

No comments

Validity of the findings

No comments

·

Basic reporting

Abstract: as the damage will not be analysed in this manuscript, I feel it should be deleted from the abstract and throughout the manuscript;

The other comments I made have been considered, so, I feel that the paper could be accepted after deleting the part of the text dealing with damage, as it is not presented in the results and not considered in the discussion.

Experimental design

OK

Validity of the findings

OK

Additional comments

OK

---

## Round 0.3 · accepted · Accept

Thank you for your revised manuscript. I have taken over as the Editor for your paper as the prior Editor is indisposed - I enjoyed reading your paper. I would particularly like to comment on the lovely illustration in Figure 1. You have addressed all of the reviewers comments and I am happy to accept it for publication.